# Correlation between Retinal Vascularization and Disease Aggressiveness in Amyotrophic Lateral Sclerosis

**DOI:** 10.3390/biomedicines10102390

**Published:** 2022-09-25

**Authors:** Gilda Cennamo, Daniela Montorio, Francesco Pio Ausiello, Luigifilippo Magno, Rosa Iodice, Alberto Mazzucco, Valentina Virginia Iuzzolino, Gianmaria Senerchia, Vincenzo Brescia Morra, Maria Nolano, Ciro Costagliola, Raffaele Dubbioso

**Affiliations:** 1Eye Clinic, Public Health Department, University of Naples Federico II, 80131 Naples, Italy; 2Eye Clinic, Department of Neurosciences, Reproductive and Odontostomatological Sciences, University of Naples Federico II, 80131 Naples, Italy; 3Clinical Neurophysiology Unit, Department of Neurosciences, Reproductive and Odontostomatological Sciences, University of Naples Federico II, 80131 Naples, Italy

**Keywords:** ALS, biomarker, vascular, disease progression, inflammation, eye, OCT, angiography, choroid, retinal nerve fiber layer

## Abstract

Abnormalities in retinal vascularization and neural density have been found in many neurodegenerative diseases; however, conflicting results are described in Amyotrophic Lateral Sclerosis (ALS). The aim of the present study was, therefore, to systematically analyze retinal layers and vascularization by means of spectral-domain (SD-OCT) and optical coherence tomography angiography (OCT-A) in ALS patients. We enrolled 48 ALS patients and 45 healthy controls. ALS patients were divided into three groups: slow progressors (n = 10), intermediate progressors (n = 24) and fast progressors (n = 14), according to the disease progression rate. For SD-OCT, we evaluated the Subfoveal choroidal thickness (SFCT), ganglion cell complex (GCC) and retinal nerve fiber layer (RNFL). Regarding the OCT-A, we assessed the vessel density (VD) in superficial and deep capillary plexuses, radial peripapillary capillary plexus, choriocapillary and the foveal avascular zone (FAZ) area. SD-OCT exam did not show any significant differences in GCC and RNFL thickness between patients and controls and among the three ALS groups. The SFCT was statistically greater in patients compared with controls (357.95 ± 55.15 µm vs. 301.3 ± 55.80 µm, *p* < 0.001); interestingly, the SFCT was thicker in patients with slow and intermediate disease progression than in those with fast disease progression (394.45 ± 53.73 µm vs. 393.09 ± 42.17 µm vs. 267.71 ± 56.24 µm, *p* < 0.001). OCT-A did not reveal any significant results. Amyotrophic Lateral Sclerosis Functional Rating Scale-Revised (ALSFRS-r) and disease duration did not correlate with any of the OCT parameters, except for SFCT with ALSFRS-r (r = 0.753, *p* = 0.024). This study demonstrated the possible association between choroidal thickness and disease activity in ALS. OCT could be a useful biomarker in the management of the disease.

## 1. Introduction

Amyotrophic Lateral Sclerosis (ALS) is the most common adult-onset neurodegenerative disease of the motor system, with a prevalence of 2–3/100,000 [1]. Despite intensive research efforts, ALS remains an incurable disease and presents with a very severe prognosis, leading to patient death within 2 to 5 years following diagnosis. ALS is characterized by the combined degeneration of both upper motor neurons located in the motor cortex and lower motor neurons located in the spinal cord [1].

The cause of this poor prognosis is the absence of a common validated etiopathogenetic theory capable of explaining the disease and, consequently, of a solving therapy. Nowadays, Riluzole is the only approved treatment, which can extend life expectancy but only by about three months [2,3,4].

Etiopathogenesis is known for the familial cases, covering about 10% of ALS cases. It is related to mutations (*SOD1, TDP43, C9orf72, FUS,* etc.) that directly promote motor neuron degradation [5]. However, recent evidence challenged the neuron-centric theory, suggesting that the dynamics of neurodegeneration in both sporadic and familial ALS are also influenced by other cell-type functions, such as glia response [6], oligodendrocyte metabolism [7] and integrity of blood vessels [8,9]. 

Specifically, central nervous system (CNS) blood vessels alterations have recently emerged as a common distinct feature in the pathogenesis of several neurodegenerative diseases, including ALS [10]. Indeed, microvascular alterations were reported in the brain and spinal cord of mice models as well as of patients’ autopsies with ALS [9]. Vascular abnormalities of CNS preceded degeneration of motoneurons and were associated with a faster disease progression and reduced survival [8]. 

Moreover, the hypothesis of vascular involvement in etiopathogenesis can be taken into consideration by evaluating the results of recent studies where abnormalities in retinal vascularization and vessel density have been found in many neurodegenerative disease [11]. Indeed, as recently demonstrated, retinal vascularization has many similar features to the cerebral vascularization [12]. In this respect, some authors suggest that metabolic demand influences, in a proportional way, both cerebral and retinal vessel density. This is greater in high-activity regions, such as the grey matter or posterior pole. Furthermore, cerebral and retinal microvasculature, due to common anatomical features, have a barrier function and shared self-regulating flow mechanisms [12]. Interestingly, it is possible to study, in vivo, retinal and choroidal vascularization by using OCT angiography (OCT-A), a non-invasive imaging technique that does not involve the injection of contrast agents but uses the blood flow inside the eye [13]. 

A better understanding of the changes in features of the retinal and choroidal blood flow could impact our ability to manage ALS and lead to new insights related to underlying pathogenesis and, potentially, indicate new therapeutic targets.

Therefore, the main aim of this cross-sectional study is to evaluate retinal and choroidal vascular plexuses in ALS patients, classified according to disease aggressiveness, using OCT in order to shed light on the microvascular abnormalities underlying this disease.

## 2. Materials and Methods

### 2.1. Participants

Forty-eight patients (96 eyes) with ALS were enrolled at the Neurology Department of the University of Naples “Federico II” from January to December 2021.

Patients met the “probable”, “probable laboratory-supported” or “definite” diagnostic categories as per the revised El Escorial criteria [14] as well as the more recent Gold Coast Criteria for ALS [15]. Genetic analysis was performed in all patients, exploring *C9orf72* repeat expansion and mutations in *SOD1*, *TARDBP* and *FUS* genes.

Patients with history of neurologic disorders affecting cognition (major stroke, severe head injuries, mental retardation), alcohol dependence or drug dependence, severe mental illness or use of high-dose psychoactive medications were not included in data analysis. Moreover, we excluded patients with diseases potentially affecting microcirculation such as diabetes, autoimmune, neoplastic or infectious diseases. 

We calculated diagnostic delay as the time interval between the time of symptom onset and date of diagnosis (in months) and the progression rate as ΔFS = (48 − Amyotrophic Lateral Sclerosis Functional Rating Scale Revised (ALSFRS-r) score at diagnosis)/diagnostic delay [16]. Patients were divided into three groups according to their ΔFS and based on a previous study [17]. Specifically, patients with ΔFS less than 0.32 (<25th percentile) represented group 1 of slow progressors, ΔFS between and including 0.32 and 0.97 (between 25th and 75th percentile) were group 2 of intermediate progressors and, lastly, patients with ΔFS more than 0.97 (>75th percentile) were group 3 of fast progressors.

Forty-five healthy subjects matched for age and sex, without any ophthalmological and neurological diseases, were recruited as control group.

Each participant underwent a complete ophthalmological examination including evaluation of best-corrected visual acuity (BCVA) according to Early Treatment of Diabetic Retinopathy Study (ETDRS), slit-lamp biomicroscopy, fundus examination with a +90 D lens, spectral domain (SD)-OCT and OCT angiography (OCT-A).

Exclusion criteria included congenital eye disorders, myopia greater than 6 dioptres, history of ocular surgery, significant lens opacities or any macular disease, previous diagnosis of glaucoma, vitreoretinal disease, uveitis and diabetic retinopathy, history of other neurological or psychiatric disorders and low-quality images obtained with OCT.

The study was approved by the Institutional Review Board of the University of Naples “Federico II” (N: 100/17/ES01) and all investigations adhered to the tenets of the Declaration of Helsinki. Signed informed consent was obtained from each subject.

### 2.2. Spectral-Domain Optical Coherence Tomography (SD-OCT)

Spectral domain-OCT (software RTVue XR version 2018.1.1.60, Optovue Inc., Fremont, CA, USA) calculated retinal nerve fiber layer (RNFL) and ganglion cell complex (GCC) thickness. The circumpapillary RNFL was calculated by the optic nerve head map protocol using a 3.45 mm radius ring centered on the optic disc. The GCC thickness was evaluated in macular region by the scan 1-mm temporal to the fovea and covering a 7 × 7 mm^2^ area [18].

### 2.3. Subfoveal Choroidal Thickness Measurement (SFCT)

The Subfoveal Choroidal Thickness (SFCT) was evaluated by means of the SD-OCT (Spectralis, Heidelberg Engineering, Heidelberg, Germany). It was measured in the subfoveal region as a manual linear measurement between the outer border of Bruch’s membrane and the most posterior identifiable aspect of the choroidal–scleral interface [19].

### 2.4. Optical Coherence Tomography Angiography (OCT-A)

Optovue Angiovue System (software ReVue XR version 2018.1.1.60, Optovue Inc., Fremont, CA, USA), based on a split-spectrum amplitude de-correlation algorithm, evaluated OCT-A images [20].

The retinal vascular networks (superficial and deep capillary plexuses) and choriocapillaris were evaluated in a 6 × 6 mm^2^ scan centered on the fovea and the VD, the percentage area occupied by the vessels in the analyzed region, was automatically calculated by AngioAnalytics™ software [21].

Angiovue software automatically calculated the foveal avascular zone (FAZ) area in square millimeters over the 6 mm x 6 mm macular area in the full retinal plexus. The “non-flow” option was selected from a drop-down menu and the area was automatically selected when the FAZ area was clicked.

The Angio-Vue disc mode analyzed the whole papillary region with a scanning area of 4.5 × 4.5 mm^2^ centered on the optic disc. Moreover, it automatically calculated the VD of radial peripapillary capillary plexus in the superficial retinal layers and extended from the inner-layer membrane to the retinal nerve fiber layer posterior boundary [22]. 

The OCT-A device included a 3D Projection Artifact Removal algorithm to remove projection artifacts, for improving depth resolution on OCT-A signal and then distinguishing vascular plexus-specific features. 

OCT-A images with a signal strength index less than 80 and residual motion artifacts were excluded from the analysis.

### 2.5. Statistical Analysis

All statistical analyses used the Statistical Package for Social Sciences (IBM SPSS Statistics, Version 25 for Windows; SPSS Inc, Chicago, IL, USA). The Shapiro–Wilk test confirmed that all variables were normally distributed. Moreover, we graphically checked the distribution of our measures by means of violin plots (Appendix A).

Chi-squared test was used to determine sex differences among groups. Linear mixed models, including age, sex and disease duration as covariates, were used to evaluate VD differences in each retinal vascular network (superficial capillary plexus, deep capillary plexus, radial peripapillary capillary plexus), in choriocapillaris and in FAZ area, using group as factor of interest. The same model was used to analyze differences in structural OCT parameters (ganglion cell complex average, retinal nerve fiber layer average and SFCT) among the groups.

Subject was included in all models as random factor to account for within-subject inter-eye correlation. 

Finally, we performed the correlations between Amyotrophic Lateral Sclerosis Functional Rating Scale revised (ALSFRS-r), disease duration and both structural OCT and OCT-A parameters using Pearson’s correlation. All correlations were graphically represented by using a matrix form with color coding based on strength and sign of correlations (i.e., red for positive correlations and blue for negative correlations).

A *p* value < 0.05 was considered statistically significant.

## 3. Results

### 3.1. Demographic and Clinical Features

Forty-eight patients (96 eyes) with ALS (15 females, 33 males; mean age 59.7 ± 11 years) and forty-five healthy controls (20 females, 25 males; mean age 58.9 ± 10 years) were analyzed. 

Group 1 included 10 patients (20 eyes) (10 males; mean age 60.2 ± 5 years), group 2 included 24 patients (48 eyes) (11 females, 13 males; mean age 61.6 ± 10 years) and group 3 included 14 patients (28 eyes) (4 females, 10 males; mean age 56 ± 15 years). Genetic analysis was negative in all patients but two, the first one harboring *C9orf72* expansion and belonging to the intermediate progressor group and the second one a *SOD1* patient belonging to the fast-progressor group.

No significant difference was found in terms of sex, age and visual acuity among the groups. Demographic and clinical features are summarized in Table 1 and Table 2. 

### 3.2. SD-OCT and OCT-A Findings

At SD-OCT exam, GCC and RNFL were not different between patients and controls, similarly among the three ALS groups (Table 1 and Table 3). 

The SFCT revealed a statistically significant increase in ALS patients with respect to controls (357.95 ± 55.15 µm vs. 301.3 ± 55.80 µm, *p* < 0.001) (Table 1); the comparison among the three patient subgroups disclosed that subfoveal choroid appeared to be thicker in patients with slow and intermediate disease progression than in those with fast disease progression (394.45 ± 53.73 µm and 393.09 ± 42.17 µm vs. 267.71 ± 56.24 µm, both *p* < 0.001), shown in Table 3 and Figure 1. In addition, the ALS fast progressor subgroup (group 3) also showed lower SFCT values compared with healthy controls, although this comparison was not statistically significant (*p* = 0.066).

The OCT-A evaluation showed that the VD in macular vascular networks (superficial and deep vascular networks) did not differ between ALS subjects and controls and among the three ALS subgroups. Further, choriocapillaris, FAZ area and radial peripapillary capillary plexus did not show any statistically significant difference between ALS subjects and controls and among the three subgroups (Table 2 and Table 3, Figure 2). 

To improve the accuracy in the classification of the data, we also performed k-means clustering that confirmed a cluster separability in the three subgroups (Appendix A).

ALSFRS-r and disease duration did not correlate with OCT-A and GCC, RNFL parameters, whereas the SFCT showed a significant positive relationship with ALSFRS-r (r = 0.753, *p* = 0.024) (Table 4).

## 4. Discussion

To our best knowledge, this is the first study that evaluates, using OCT-A, the characteristics of the retinal and choroidal vascular plexuses in ALS patients with different disease aggressiveness. 

In the past few years, several OCT studies assessed retinal nerve fibers as well as vasculature involvement in ALS patients. Indeed, some authors focused on the analysis of the state of the optic nerve by measuring the thickness of RNFL and conflicting observations have emerged. The initial evidence that suggested a thinning of the RNFL was subsequently countered by other evidence with the opposite orientation [23]. Specifically, some studies found a significant thinning of the RNFL in ALS patients without previous known ophthalmic diseases [24,25,26,27]. On the other side, other authors [28,29,30], including larger ALS samples [28], showed no significant alterations in RNFL. Similar conflicting results are also described for the other retinal layers [23]. In detail, while few works described no significant changes, three studies have shown that it is possible to detect modifications in them [23]: two studies reported alterations in the thickness of the INL [24,30] and the other demonstrated a significant thinning of the ONL [29]. However, none of the mentioned studies correlated ocular findings with the disease progression rate. 

Another aspect assessed in the literature is related to abnormalities in retinal vasculature. The unique significant information in this regard relates to changes in vessel thickness. As previously shown for dermal blood vessels and muscular capillaries [31,32,33], Abdelhak et al. further found a thickening of the vessel walls in the retina [29]. It has been postulated, because of similarities with systemic vessels, that this is a consequence of the duplication of basement membranes or the deposition of amyloid B-protein [34]. The latter cause would induce narrowing of blood vessels through inflammatory vasculopathy. Similar alterations have been found in the brain and spinal cord in mouse models of ALS [8]. In this scenario, we developed this study. Our results, in fact, confirmed the absence of statistically significant variations in the GCC and RNFL values between patients and controls and among the three ALS groups. This trend was the same for choriocapillaris, FAZ area and radial peripapillary capillary plexus. On the contrary, an interesting result comes from the evaluation of the SFCT, which resulted to be increased in patients compared to controls. It is also remarkable that the SFCT was different among the three patient subgroups. It appeared to be thicker in patients with slow and intermediate disease progression than in those with fast disease progression.

We hypothesized that patients with a slow and intermediate disease progression may display a vasculopathy determining an increased leakage of inflammatory cells in the parenchyma of choroid with a consequent increase in SFCT. Conversely, for fast progressors, we speculated that a different and more destructive inflammatory profile [35] may thin the choroid, because of massive insult to the microvasculature with a subsequent atrophy. Indeed, choroid plexus plays a pivotal role in the recruitment of immunoregulatory cells to the central nervous system [36] and such “immunological filtering” can determine the different disease aggressiveness and, therefore, distinct survival trajectories among ALS patients. 

Notably, T cells, monocytes and other peripheral immune cells can directly access and/or indirectly influence the central nervous system via the choroid plexus [37]. T cell infiltration to the CNS has for a long time been linked only to severe pathology, but it is now evident that CD4 T cells, in particular Tregs, also exert an important neuroprotective function. Therefore, the theory that disease progression in ALS is shaped by immune factors is now widely accepted [35]; indeed, higher levels of IL-6 and IL-6R or reduced Treg numbers and function are linked to rapid disease progression [35]. 

During inflammation, the choroidal thickness may change. Indeed, patients with multiple sclerosis, especially those with optic neuritis, display a thinner choroid compared with healthy controls, suggesting the role of active and destructive inflammation in the choroidal vascular alterations [38]. Conversely, choroidal thickness might also increase due to the occurrence of neovascularization phenomena secondary to inflammatory insults [39]. Interestingly, preclinical data show that in ALS, angiogenesis can help prevent oxidative stress and, hence, promote cell survival [40].

Therefore, we can speculate that choroidal thickness in ALS patients could reflect the different intensity of inflammation, which results in a different degree of neurodegeneration and vascularization, mirroring disease aggressiveness. 

In addition, the alteration in choroidal thickness did not affect the blood flow signal detected at the OCT-A and, for this reason, the choriocapillary VD did not show any significant change in different disease stages. The latter finding might be explained by the fact that in ALS, there is a selective involvement of the larger choroidal vessels, belonging to the external and medium layers and keener to inflammatory insult coming from the systemic circulation. Importantly, the integrity of inner layers, represented by choriocapillaris vessels, might also explain the relative preservation of retinal layers. This result is of importance, since in other neurodegenerative diseases, retinal layers are usually affected early in the disease course. 

The increased inflammatory vasculopathy was also confirmed by a previous study that reported a significant alteration in cerebral vascular integrity in choroid plexus of ALS patients, resulting in structural and functional disruptions to the blood–cerebrospinal fluid barrier, without defining any association with disease activity or progression, however [41]. 

Our findings strengthen the similarity between ocular and cerebral vasculature. Indeed, choroidal vasculature supports metabolic demand of retinal pigment epithelium and photoreceptors [42]. Analogously, the choroid plexus, localized within ventricles, produces cerebrospinal fluid that provides protection and nourishment to the brain [43].

The choroidal vasculature is becoming a crucial factor for understanding the pathogenesis and improving the follow-up of neurodegenerative diseases.

Indeed, the use of retinal scanning, especially OCT-A, to quantify retinal vascular network abnormalities, including choroidal vasculature, is a promising tool to evaluate disease severity in numerous neurodegenerative disorders, such as multiple sclerosis, Parkinson’s disease, Alzheimer’s disease, and Huntington’s disease [11]. Interestingly, dysregulation of the choroid plexus, which shares functional and morphological similarities with the choroidal vasculature [12], reflects a common underlying mechanism in the pathophysiology of such neurodegenerative diseases, since it represents a unique neuro-immunological interface, positioned to integrate signals it receives from the cerebral parenchyma with signals coming from circulating immune cells [44].

The results of our study, considering the similarities between retinal and cerebral vascularization [12], could lead to a search for an etiopathogenetic theory of the disease based on vascular alterations, as an alternative perspective to the classic neuron-centric hypothesis. Furthermore, OCT is a simple and non-invasive tool that can be easily repeated over time and, therefore, could be potentially used as a biomarker to evaluate the efficacy of new therapies targeting brain microcirculation to improve the prognosis of these patients.

There are some shortcomings in our research, which should be considered. First, even if the ALS fast progressor subgroup had the lower SFCT value compared to healthy controls, such a difference was not statistically significant. We think that a significant result could likely be evident only by recruiting more ALS patients in each subgroup. Second, the cross-sectional design of our study may have limited our findings; therefore, future longitudinal studies are needed to confirm the results of the present study. Third, the lack of inflammatory profile estimation in patients and controls does not allow one to directly relate retinal vascular change to inflammatory process. However, studies in multiple sclerosis suggested that the impaired retinal microcirculation could be a consequence of direct inflammatory damage to the vessels [45].

## 5. Conclusions

This study demonstrated the possible association between SFCT value and disease activity in ALS, suggesting that this parameter might be a new possible biomarker of disease aggressiveness and treatment response.

## Figures and Tables

**Figure 1 biomedicines-10-02390-f001:**
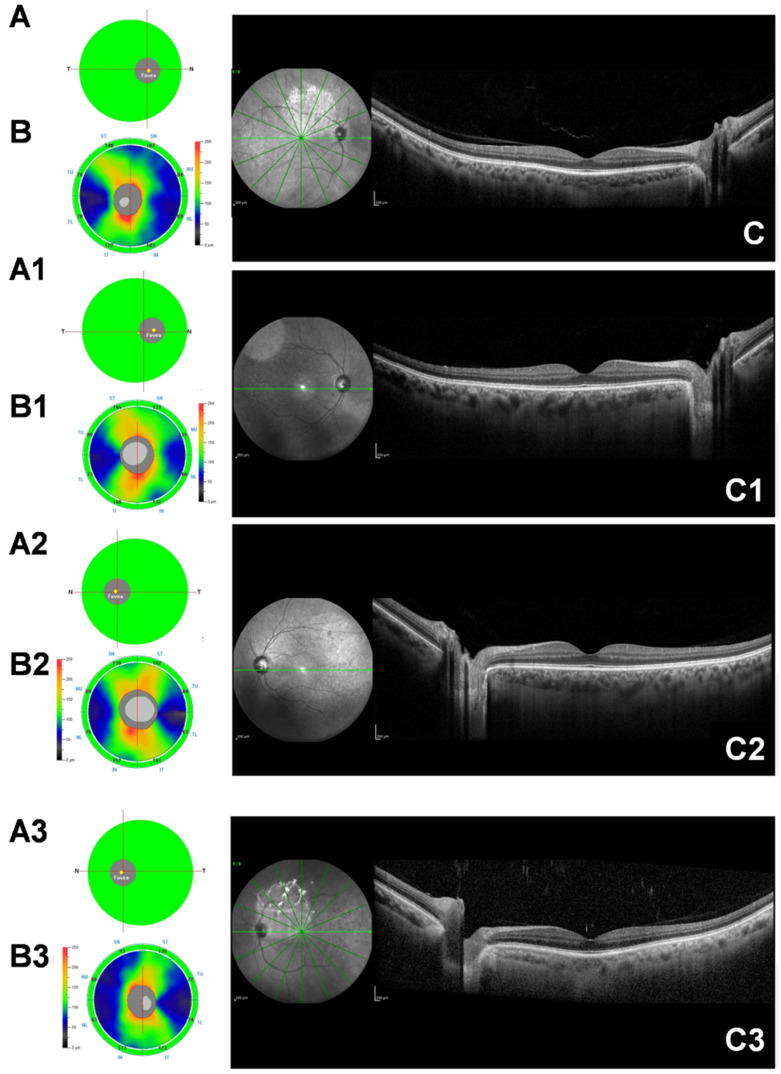
Spectral domain optical coherence tomography (SD-OCT) images from a healthy subject’s right eye (male, 58 years) with normal thicknesses of ganglion cell complex (GCC) (**A**) and retinal nerve fiber layer (RNFL) (**B**). The OCT B-scan shows normal retinal and choroidal thickness (**C**). Patient’s right eye (female, 60 years) affected by amyotrophic lateral sclerosis (ALS) with slow disease progression. SD-OCT shows normal GCC and RNFL thickness (**A1**,**B1**). The OCT B-scan shows normal retinal thickness and a thicker choroid (**C1**). Patient’s right eye (female, 61 years) affected by ALS with intermediate disease progression. SD-OCT shows normal GCC and RNFL thickness (**A2**,**B2**). The OCT B-scan shows normal retinal thickness and an increased choroidal thickness (**C2**). The bottom row reveals a patient’s left eye (female, 57 years) affected by ALS with fast disease progression. SD-OCT shows normal GCC and RNFL thickness (**A3**,**B3**). The OCT B-scan shows normal retinal and a thinner choroid (**C3**).

**Figure 2 biomedicines-10-02390-f002:**
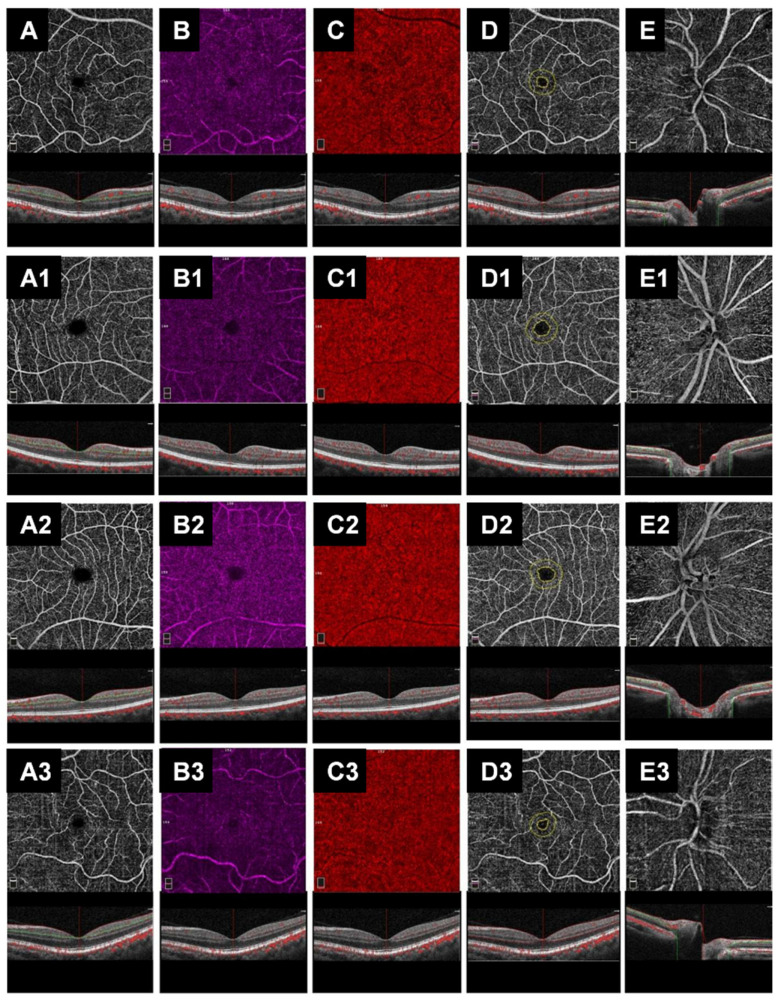
OCT-A from a healthy subject’s right eye (male, 58 years) shows normal vasculature texture in macular region (retinal superficial, deep capillary plexuses and in coriocapillary) and in papillary region (**A**–**C**,**E**). The foveal avascular zone (FAZ) area shows a normal size (**D**). The second row shows the patient’s right eye (female, 60 years) affected by amyotrophic lateral sclerosis (ALS) with slow disease progression. OCT-A images reveal absence of abnormalities in retinal and choriocapillary vascular plexuses in macular and papillary regions (**A1**,**B1**,**C1**,**E1**). The FAZ shows a normal area (**D1**). The third row shows a patient’s left eye (female, 61 years) affected by ALS with intermediate disease progression. OCT-A reveals absence of abnormalities in retinal and choriocapillary vascular plexuses in macular and papillary regions (**A2**,**B2**,**C2**,**E2**). The FAZ shows a normal area (**D2**). The bottom row reveals the patient’s left eye (female, 57 years) affected by ALS with fast disease progression. OCT-A images show normal retinal and choriocapillary vascular plexuses in macular and papillary regions (**A3**,**B3**,**C3**,**E3**). The FAZ shows a normal area (**D3**).

**Table 1 biomedicines-10-02390-t001:** Demographic and clinical characteristics of ALS patients and healthy controls.

	Controls	ALS Patients	*p* Value
Eyes (N.)	90	96	-
Age (years)	58.9 ± 10	59.7 ± 11	
Sex (female/male)	20/25	15/33	
ALSFRS-r	-	36 ± 3.6	-
ΔFS score	-	0.64 ± 0.63	-
Disease duration (months)	-	40.2 ± 40	-
Diagnostic delay (months)		26.5 ± 29.1	
OCT-A parameters (%)			
*SCP* Whole	48.81 ± 3.44	48.26 ± 4.34	0.875
*DCP* Whole	49.45 ± 4.92	50.10 ± 5.47	0.764
*CC* Whole	71.59 ± 4.83	70.27 ± 4.55	0.768
*RPC* Whole	48.57 ± 2.99	47.70 ± 3.12	0.896
*FAZ* area (mm^2^)	0.278 ± 0.08	0.281 ± 0.07	0.922
SD-OCT parameters (µm)			
*GCC* average	98.45 ± 7.01	98.37 ± 6.80	0.914
*RNFL* average	100.9 ± 9.63	101.74 ± 8.32	0.862
*SFCT* average	301.3 ± 55.80	357.94 ± 55.15	<0.001
BCVA (logMAR)	0.06 ± 0.05	0.07 ± 0.03	0.789

ALSFRS-r: Amyotrophic Lateral Sclerosis Functional Rating Scale Revised; ΔFS score= rate of disease progression; OCT-A: Optical Coherence Tomography Angiography; SD-OCT: spectral domain OCT; SCP: Superficial Capillary Plexus; DCP: Deep Capillary Plexus; CC: Choriocapillaris; RPC: Radial Peripapillary Capillary; FAZ: Foveal Avascular Zone; GCC: Ganglion Cell Complex; RNFL: Retinal Nerve Fiber Layer; SFCT: Subfoveal Choroidal Thickness; BCVA: best-corrected visual acuity; logMAR: logarithm of the minimum angle of resolution. Data are expressed as mean ± standard deviation.

**Table 2 biomedicines-10-02390-t002:** Demographic and clinical characteristics of ALS groups.

	Group 1	Group 2	Group 3
Eyes (N.)	20	48	28
Age (years)	60.2 ± 5	61.6 ± 10	56 ± 15
Sex (female/male)	-/10	11/13	4/10
ALSFRS-r	36.5 ± 4.2	32.1 ± 5	33.7 ± 4.3
ΔFS score	0.13 ± 0.04	0.48 ± 0.18	1.43 ± 0.73
Disease duration (months)	96.3 ± 65	37 ± 15	11.6 ± 5
Diagnostic delay (months)	54.15 ± 44.66	21.4 ± 10.3	8.3 ± 4.0
OCT-A parameters (%)			
*SCP* Whole	48.80 ± 4.26	48.26 ± 4.15	46.8 ± 6.55
*DCP* Whole	50.58 ± 6.60	48.93 ± 7.08	48.7 ± 7.44
*CC* Whole	70.95 ± 6.94	70.03 ± 3.16	68.7 ± 6.44
*RPC* Whole	47.53 ± 2.63	46.11 ± 3.44	47.5 ± 5.14
*FAZ* area (mm^2^)	0.279 ± 0.08	0.280 ± 0.12	0.281 ± 0.11
OCT parameters (µm)			
*GCC* average	98.45 ± 6.93	97.25 ± 11.54	95.1 ± 10.33
*RNFL* average	101.07 ± 8.44	99.51 ± 10.38	98.5 ± 12.25
*SFCT* average	394.45 ± 53.73	393.09 ± 42.17	267.71 ± 56.24
BCVA (logMAR)	0.07 ± 0.04	0.01 ± 0.03	0.03 ± 0.05

ALSFRS-r: Amyotrophic Lateral Sclerosis Functional Rating Scale Revised; ΔFS score= rate of disease progression; OCT-A: Optical Coherence Tomography Angiography; SCP: Superficial Capillary Plexus; DCP: Deep Capillary Plexus; CC: Choriocapillaris; RPC: Radial Peripapillary Capillary; FAZ: Foveal Avascular Zone; GCC: Ganglion Cell Complex; RNFL: Retinal Nerve Fiber Layer; SFCT: Subfoveal Choroidal Thickness; BCVA: best-corrected visual acuity; logMAR: logarithm of the minimum angle of resolution. Data expressed as mean ± standard deviation. Group 1: ALS slow progressors; Group 2: ALS intermediate progressors; Group 3: ALS fast progressors.

**Table 3 biomedicines-10-02390-t003:** Differences in SD-OCT and OCT-A parameters among ALS subgroups.

	Group 1 vs. Group 2	
SD-OCT	β	*p*-Value
GCC average	1.445	0.622
RNFL average	2.371	0.694
SFCT average	1.321	0.824
OCT-A		
SCP Whole	0.130	0.784
DCP Whole	2.142	0.344
CC Whole	0.240	0.633
RPC Whole	1.342	0.405
FAZ area	−0.011	0.626
	**Group 1 vs. Group 3**	
**SD-OCT**	**β**	***p*-Value**
GCC average	3.621	0.514
RNFL average	3.469	0.531
SFCT average	126.32	<0.001
OCT-A		
SCP Whole	2.417	0.534
DCP Whole	2.342	0.426
CC Whole	2.602	0.524
RPC Whole	−0.047	0.726
FAZ area	−0.019	0.844
	**Group 2 vs. Group 3**	
**SD-OCT**	**β**	***p*-Value**
GCC average	2.134	0.663
RNFL average	1.247	0.752
SFCT average	125.41	<0.001
OCT-A		
SCP Whole	2.345	0.431
DCP Whole	0.034	0.741
CC Whole	2.311	0.524
RPC Whole	−1.674	0.423
FAZ area	−0.015	0.647

GCC: Ganglion cell complex; RNFL: Retinal nerve fiber layer; SFCT: Subfoveal choroidal thickness SCP: Superficial Capillary Plexus; DCP: Deep Capillary Plexus; CC: Choriocapillaris; RPC: Radial Peripapillary Capillary; FAZ: Foveal avascular zone. Group 1: ALS slow progressors; Group 2: ALS intermediate progressors; Group 3: ALS fast progressors.

**Table 4 biomedicines-10-02390-t004:** Pearson’s correlations between OCT measures and clinical findings represented as visual matrix form with color coding based on strength and sign of correlations (shades of red for positive correlations and shades of blue for negative correlations). Please note the positive significant relationship between ALSFRS-r and subfoveal choroidal thickness (SFCT).

	Disease Duration	ALSFRS-r	SCP	DCP	CC	RPC	FAZ	GCC	RNFL	SFCT
**Disease Duration**	1									
**ALSFRS-r**	−0.1149	1								
**SCP**	0.0707	0.1413	1							
**DCP**	−0.1326	0.2561	0.3858	1						
**CC**	−0.1048	0.0976	0.2002	0.1941	1					
**RPC**	−0.0103	0.0760	0.4693	0.0301	0.2016	1				
**FAZ**	0.2218	−0.2739	−0.1753	0.0719	−0.0130	−0.0080	1			
**GCC**	0.0843	0.2179	0.4119	0.1434	0.1148	0.3980	−0.1272	1		
**RNFL**	0.0701	0.2469	0.3771	0.3122	0.1507	0.4082	0.1271	0.6505	1	
**SFCT**	0.2443	0.7526	0.0728	0.0827	0.0472	−0.1976	−0.0603	0.0754	0.0009	1

## Data Availability

Data are available upon request.

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
