# Peer review of "Correlation between Retinal Vascularization and Disease Aggressiveness in Amyotrophic Lateral Sclerosis"

_biomedicines, 2022, doi:10.3390/biomedicines10102390_

Round 1

Reviewer 1 Report

In this study, Dubbioso and colleagues investigated the association between retinal vascularization and disease aggressiveness in Amyotrophic Lateral Sclerosis (ALS) using spectral-domain (SD-OCT) and Optical Coherence Tomography angiography (OCT-A). They found that Subfoveal Choroidal Thickness (SFCT) could be a potential biomarker to indicate disease activity in ALS. The study design and applied methods are adequate to address the research questions. However, I have some concerns on the conclusion.

Mayor comments:

1.      1.    According to the results, SFCT average value in the ALS patient group (357.94±55.15) is significantly higher than healthy controls (301.3±55.8), while the ALS fast progressors subgroup (group 3) has a lower SFCT value (267.71±56.24) than healthy controls.  Has the difference between Group 3 and the control group been statistically tested? If the difference is significant, the authors must carefully infer that there is a link between choroidal thickness and disease activity, or at the very least disclose the study's limitations.

2.       2.  The authors proposed an inflammatory vascular phenomenon hypothesis to explain the observed differences; however no studies were undertaken to corroborate this. They could discuss it in the discussion section, but they couldn’t make conclusions from it.

Minor comments:

1.   Do the ALS participants belong to any genetic ALS cases.

2.   Typo in line 308, “SLA” should be “ALS.”

Author Response

Response to Reviewer 1 Comments 

Major comments:

Point 1: According to the results, SFCT average value in the ALS patient group (357.94±55.15) is significantly higher than healthy controls (301.3±55.8), while the ALS fast progressors subgroup (group 3) has a lower SFCT value (267.71±56.24) than healthy controls.  Has the difference between Group 3 and the control group been statistically tested? If the difference is significant, the authors must carefully infer that there is a link between choroidal thickness and disease activity, or at the very least disclose the study's limitations.

Response 1: We thank the reviewer for raising this important point. The difference between the ALS fast progressors subgroup (group 3) and healthy controls is not significantly different (p= 0.066), even if the Group 3 displayed a lower value. We reason that such difference would become significant if we had recruited more patients with fast-progressing disease. We have now added such issue as limitation of our study, please see the discussion section (page 13, lines 361-370).

There are some main shortcomings of our research to be considered. First, even if the ALS fast progressors subgroup had a lower SFCT value than healthy controls, such difference was not statistically significant. We think that a significant result could likely be evident only recruiting more ALS patients in each subgroup. Second, the cross-sectional design of our study may have limited our findings, therefore future longitudinal studies are needed to confirm the results of the present study. Third, the lack of inflammatory profile estimation in patients and controls does not allow to directly relate retinal vascular change to inflammatory process. However, studies in multiple sclerosis suggested that the impaired retinal microcirculation could be a consequence of direct inflammatory damage to the vessels [45].”

Point 2: The authors proposed an inflammatory vascular phenomenon hypothesis to explain the observed differences; however, no studies were undertaken to corroborate this. They could discuss it in the discussion section, but they couldn’t make conclusions from it.

Response 2: We thank the reviewer for the comment. We have now toned down in the conclusion of the manuscript the inflammatory vascular hypothesis and at the same time we have better explain in the discussion section the relation between choroid thickness changes and the inflammatory hypothesis (page 12).

Minor comments:

Point 1: Do the ALS participants belong to any genetic ALS cases.

Response 1: We thank the reviewer for alerting us to this point. We also recruited two genetic ALS cases: the first one harbouring the C9orf72 expansion and belonging to the intermediate progressor group and the second one a SOD1 patient belonging to the fast-progressor group. We have now specified in the text this info (page 4, lines 177-180). 

“Genetic analysis was negative in all patients but two, the first one harbouring the C9orf72 expansion and belonging to the intermediate progressor group and the second one a SOD1 patient belonging to the fast-progressor group”

Point 2: Typo in line 308, “SLA” should be “ALS.”

Response 2: Done.

Reviewer 2 Report

An interesting study with OCT and OCT-A. The text is relatively difficult to read - many abbreviations and specific method - OCT targetting vessels. I have comments:

1.El Escorial criteria - why not some more modern critaria?

2.Inflammatory changes of choroid vessels- it would be better to describe more inflammatory changes of vessels and to discuss this changes.

3.The alterations of choroid thicness - did not affect the blood flow - no changes in retinal and choroid vessel density. So the effect of vascular changes and perfusion changes in patients with various grades of ALS  progression should be better defined. 

Author Response

Response to Reviewer 2 Comments

Point 1: El Escorial criteria - why not some more modern criteria?

Response 1: We thank the reviewer for raising this point. We have now added in the text that our patients also met the new Gold Coast criteria for ALS (Shefner et al., 2020), page 2, lines 87-90.

Patients met the “probable”, “probable laboratory-supported” or “definite” diagnostic categories as per the revised El Escorial criteria [14] as well as the more recent Gold Coast Criteria for ALS [15].  Genetic analysis was performed in all patients, exploring C9orf72 repeat expansion and mutations of SOD1, TARDBP and FUS genes.

Point 2: Inflammatory changes of choroid vessels- it would be better to describe more inflammatory changes of vessels and to discuss these changes.

Response 2: We thank the reviewer for the suggestion. We have now better explained in the discussion the relationship between choroid thickness change and inflammatory hypothesis related to the disease severity (page 12).

Point 3: The alterations of choroid thickness - did not affect the blood flow - no changes in retinal and choroid vessel density. So, the effect of vascular changes and perfusion changes in patients with various grades of ALS progression should be better defined. 

Response 3: We thank the reviewer for raising this important point. We have now better explained in the discussion that in ALS there might be a selective involvement of the larger choroidal vessels, belonging to the external and medium choroid layers and keener to inflammatory insult coming from the systemic circulation. On the contrary, the inner layers (i.e., choriocapillaris layer) seems to be unaffected and such results could also explain the relative preservation of the retinal layers (page 12, lines 325-332).

In addition, the alteration of choroidal thickness did not affect the blood flow signal detected at the OCT-A, and for this reason the choriocapillary VD did not show any significant change in different disease stages. The latter finding might be explained by the fact that in ALS there is a selective involvement of the larger choroidal vessels, belonging to the external and medium layers and keener to inflammatory insult coming from the systemic circulation. Importantly, the integrity of inner layers, represented by choriocapillaris vessels, might also explain the relative preservation of retinal layers. This result is of importance, since in other neurodegenerative disease retinal layers are usually early affected over the disease course.

Reviewer 3 Report

The authors present an interesting study examining the effects of retinal vascularization in ALS.  The topic and content is important and relatively novel to the field.  The clinical collection of parameters and overall presentation is sound. However, there are a few areas of improvement necessary in the statistical analysis and corresponding results presentation.

The authors choose linear mixed effects model after finding the data met the Shapiro Wilks requirement. However, beyond the beta values and p-value, there is little to judge the goodness of fit to check for things like multicollinearity, interaction terms, overfitting, underfitting, etc.  This information can be better assessed graphically (currently no visualizations are shown) and with additional data views provided in the appendix or supplement (raw data versus model, etc.). Violin plots with raw data would give a better feel of any possible skew that might adversely impact the analysis if not corrected by the authors in preprocessing.

Moreover, while linear models are often used in ALS, the linear assumptions do not always hold.  It would help helpful to do simple paired test of key retinal metrics between groups (with multiple comparison correction factor like Bonferroni or Tukey, where applicable) to prevent type I error.

More info on the exact SPSS packages are necessary to infer both implicit and explicit parameters impacting the analysis.

No details were given about data preprocessing (normalization or standardization, etc.) which can definitely impact model fits or what the authors did with missing data (if present).

The Pearson's correlations should be provided in visual matrix form with color coding based on strength and sign.

Another recommended assessment would be to perform clustering (even something like k-means with dimensional reduction) using these features to see if there is cluster separability based on specific features that could provide additional clinical context.

More information is necessary to assess the categories of slow, intermediate, etc. The methods says percentiles were used. Were these percentiles based solely on the this study's sample or a larger data set? If so, there could be substantive bias.  Using clustering with progression type as a feature would be another way to help ascertain whether this somewhat arbitrary definition of progression is providing any additional context for this study.   If it's not, it would be prudent to simply compare ALS to control to quantify overall effect size of retinal metrics of interest.

Finally, the results implications in the Discussion are currently overstated based on the present statistical analysis. More analysis is needed to project the authors assumption that their original etiological hypothesis holds.  More analysis per above would be helpful.  Also, the authors may choose to simply pull back on the tone of the writing in this section.

While the idea of retinal scanning is relatively new in ALS, it has been used for several other neurological diseases or vascular-related diseases. Drawing from some those literatures for context would be helpful for the present study for readers who may be less familiar with retinal etiology or pathology.

Round 2

Reviewer 3 Report

The authors have completed all requested revisions.  The work is much improved in technical quality and presentation.